# Risk factors for mental illness in adults with atopic eczema or psoriasis: protocol for a systematic review

Elizabeth I Adesanya ![ORCID],[1] Yochai Schonmann,[2,3] Joseph F Hayes,[4] Rohini Mathur ![ORCID],[1] Amy R Mulick ![ORCID],[1] Lauren Rayner,[5] Liam Smeeth,[1] Catherine H Smith,[6] Sinéad M Langan ![ORCID],[1,7] Kathryn E Mansfield ![ORCID][1]

**Correspondence to**
Elizabeth I Adesanya;
elizabeth.adesanya@lshtm.ac.uk

## ABSTRACT

**Introduction** Evidence indicates that people with the common inflammatory skin diseases atopic eczema or psoriasis are at increased risk of mental illness. However, the reasons for the relationship between skin disease and common mental disorders (ie, depression and anxiety) or severe mental illnesses (ie, schizophrenia, bipolar disorder and other psychoses) are unclear. Therefore, we aim to synthesise the available evidence regarding the risk factors for mental illness in adults with atopic eczema or psoriasis.

**Methods and analysis** We will conduct a systematic review of randomised controlled trials, cohort, case–control and cross-sectional studies. We will search the following databases from inception to March 2020: Medline, Embase, Global Health, Scopus, the Cochrane Library, Web of Science, Base, PsycInfo, the Global Resource of Eczema Trials, and the grey literature databases Open Grey, PsycExtra and the New York Academy of Medicine Grey Literature Report. We will also search the bibliographies of eligible studies and relevant systematic reviews to identify additional relevant studies. Citation searching of large summary papers will be used to further identify relevant publications. Two reviewers will initially review study titles and abstracts for eligibility, followed by full text screening. We will extract data using a standardised data extraction form. We will assess the risk of bias of included studies using the Quality in Prognosis Studies tool. We will synthesise data narratively, and if studies are sufficiently homogenous, we will consider a meta-analysis. We will assess the quality of the evidence using the Grading of Recommendations, Assessment, Development and Evaluation framework.

**Ethics and dissemination** Ethical approval is not required for a systematic review. Results of the review will be published in a peer-reviewed journal and disseminated through conferences.

**PROSPERO registration number** CRD42020163941.

## Strengths and limitations of this study

- ► This protocol promotes transparent review methods, enables comparison of our final review to our initial plans, minimises risk of bias, and reduces the chance of unplanned duplication.
- ► Our systematic review will be the first to critically evaluate studies of the risk factors for mental illness in adults with atopic eczema or psoriasis.
- ► We will ensure our review is comprehensive by searching multiple scientific literature databases (including specific grey literature databases), including a range of study types and not limiting to English-language studies.
- ► However, the studies we include may use heterogenous methods and be of variable quality, which may limit our ability to calculate pooled estimates from meta-analysis and may limit our conclusions.

2.6% of adults,[1] and the prevalence of atopic eczema in adults is approximately 2.5%.[2] Similarly, mental illness is common. According to the 2017 Global Burden of Disease Study, mental illness is one of the leading causes of years lived with disability worldwide.[3] In England, 17% of adults have common mental disorders (CMD- including depression or anxiety).[4] Severe mental illness (SMI—including schizophrenia, bipolar affective disorder and other psychoses) affects nearly 1% of the UK population.[4] Individuals with SMI experience substantial health inequalities; they are at increased risk of serious health problems (eg, diabetes mellitus and cardiovascular disease) and die up to 20 years earlier than the general population.[4,5]

Associations between atopic eczema or psoriasis and mental illness are well established. Evidence suggests that people with atopic eczema or psoriasis are at increased risk of mental illness.[6–14] The temporal sequence of the associations between skin disease and mental illness is also well recognised, with evidence suggesting that atopic eczema or

## INTRODUCTION

Psoriasis and atopic eczema are inflammatory skin conditions associated with considerable morbidity and reduced quality of life for both sufferers and their families. Atopic eczema and psoriasis are common in the UK population—psoriasis affects between 1.3% and

psoriasis precedes mental illness diagnosis.[10 12] However, the reasons for the relationship between inflammatory skin disease and mental illness are unclear. To the best of our knowledge, there are no existing systematic reviews addressing risk factors for the relationship between atopic eczema or psoriasis and mental illness in adults. Previous systematic reviews have aimed to establish summary measure of effects for the association between either atopic eczema or psoriasis and specific mental illnesses (eg, depression); the majority have focused on the relationship between atopic eczema or psoriasis and CMDs.[15–19] One systematic review has investigated the risk factors that mediate the association between atopic eczema and mental illness in children and adolescents only. The majority of studies in this review of children were conducted in European countries or territories. Meta-analysis of the 35 studies included in the review found that although demographic factors such as age, sex and socioeconomic status did not moderate the risk of developing mental illness in children with atopic eczema, children from predominantly minority ethnic backgrounds were more likely to be diagnosed with a mental illness in comparison with their Caucasian counterparts.[20]

The primary aim of this systematic review will be to explore, synthesise and critically evaluate the strength and quality of all available evidence on the risk factors associated with the development of mental illness (CMDs and SMIs) in adults with atopic eczema or psoriasis. If possible, we will also compare and contrast the risk factors associated with the development of mental illness in adults with atopic eczema to the risk factors in psoriasis. In the context of this systematic review, we will use the term 'risk factor' to refer to variables associated with an increased risk of mental illness in individuals with atopic eczema or psoriasis.

## METHODS
This study protocol adheres to the Preferred Reporting Items for Systematic Review and Meta-Analysis Protocols (PRISMA-P).[21]

### Patient and public involvement
Patients and/or the public were not involved in this systematic review protocol.

### Eligibility criteria
We will screen studies for potential inclusion in our review according to the eligibility criteria presented in table 1.

### Information sources
We will search the following databases for relevant articles from inception to March 2020: Medline, Embase, Global Health, Scopus, the Cochrane Library (which includes Cochrane Reviews, Cochrane Protocols, Trials, Editorials, Special Collections, Clinical Answers and Other Reviews),

| Table 1 | Eligibility criteria | |
| --- | --- | --- |
| | **Inclusion criteria** | **Exclusion criteria** |
| Study design and characteristics | All RCTs, cohort, case–control and cross-sectional studies where an effect estimate (ie, ratio or difference measures) of the risk factors for mental illness in adults with atopic eczema or psoriasis is reported.<br><br>Studies in any language and from any geographical setting will be considered. | Ecological studies, case series studies, case reports and systematic reviews (however, relevant summary reviews will be flagged and reference lists searched for eligible studies).<br><br>Studies where correlates (without a measure of effect) have been calculated to estimate the association between a risk factor and mental illness in adults with atopic eczema or psoriasis.<br><br>Conference proceedings, letters, editorials, opinion articles and reports (however, relevant conference proceedings/letters will be flagged to try and identify full text). |
| Participants | Human participants aged 18 and over with atopic eczema or psoriasis.<br><br>Studies including both adults and children where data for adults is reported separately. | Studies conducted in children or adolescents only.<br><br>Animal or cell studies. |
| Exposure | Risk factors for mental illness (CMD or SMI). | |
| Comparators | Studies must compare adults with atopic eczema or psoriasis with the risk factors of interest with adults with atopic eczema or psoriasis without the risk factors of interest. | |
| Outcomes | Study outcomes must be a CMD or SMI, either clinically diagnosed or self-reported with or without validated tools. | |

CMD, common mental disorder; RCT, randomised controlled trial; SMI, severe mental illness.

**Table 2** Keywords included in the search strategy for all databases

| Search term | Keywords |
|---|---|
| Risk factor terms | risk OR risk factor* OR protective factor OR predict* OR correlat* OR associat* OR aetiol* OR etiol* OR relationship OR mediat* OR mechanism* OR caus* OR path* |
| Atopic eczema or psoriasis terms | atopic dermatitis OR eczema OR atopy OR psoriasis OR psoria* OR (pustulo* AND palmopl* OR palmari* OR palmar) |
| Mental illness terms | mental health OR mental* ill* OR mental disorder* OR psychiatr* ill* OR psychiatr* disorder OR psychiatr* disease* OR psychological* ill* OR psychological* disorder* OR psychological* disease* OR affective* OR anxi* OR depress* OR phobi* OR panic OR bipolar* OR schizophrenia OR schizo* OR delusion* OR psychotic* OR psychos#s OR psychological* distress |

Web of Science (which includes the Science Citation Index Expanded, the Social Sciences Citation Index, the Arts & Humanities Citation Index, the Conference Proceedings Citation Index-Science, the Conference Proceedings Citation Index—Social Science & Humanities and the Emerging Sources Citation Index), Base, PsycInfo and the Global Resource of Eczema Trials. Both Medline and Embase capture a large amount of published literature—Medline indexes more than 5200 journals, and Embase indexes almost 8500 journals[22 23]—while the other databases are likely to contain appropriate papers for this review. To ensure that all relevant literature is included in the review, we will also search for grey literature in Open Grey, the New York Academy of Medicine Grey Literature Report and PsycExtra. Finally, we will search the five largest clinical trial registries—ClinicalTrials.gov, the EU Clinical Trials Register, the Japan Primary Registries Network, International standard Randomised Controlled Trial Number (ISRCTN) and the Australian New Zealand Clinical Trials Registry—to identify relevant trials.[24]

### Search strategy
We will search medical subject headings and free text (in titles, abstracts and keywords) for synonyms relating to three key concepts: (1) 'risk factors', (2) 'atopic eczema or psoriasis' and (3) 'mental illness' (table 2). We will combine the three key concepts in the search strategy using the Boolean logic operator 'AND'. We have developed and piloted an initial search strategy in the Medline database that has been peer reviewed by a librarian (online supplemental table 1), and we will adapt it appropriately for other databases. We will also manually scrutinise the reference lists and bibliographies of relevant systematic reviews to identify additional papers for inclusion. Finally,

we will use citation searching on large summary papers to identify any further relevant publications.

### Study records
#### Data management
A single reviewer (EA) will import all results returned from the electronic database searches into the reference management tool EndNote V.X9 (Clarivate Analytics, V.9.2/2019). After identifying and removing duplicate records, we will import the search results into Rayyan (a web application for systematic reviews),[25] where the integrated deduplication function will be used to identify any previously missed duplicates.

#### Study selection
Two reviewers (EA and YS) will independently screen the titles and abstracts of the search results for potentially relevant studies. Both reviewers will then screen the full text of all potentially relevant studies for inclusion using the eligibility criteria. Any disagreements during this process will be discussed by EA and YS, with consultation from a third reviewer (KM) if necessary. We will record and report in a flowchart the reasons for study rejection following full text screening.

#### Data extraction
We will develop a standardised data extraction form (to extract information described below), which will be piloted by two reviewers (EA and YS) who will extract data from the larger of either 10% or five of the eligible studies. Any disagreements between the two reviewers will be discussed, with a third reviewer (KM) available to arbitrate if required, and changes made to the data extraction form if necessary. A single reviewer (EA) will complete the extraction of data for the remaining studies. We will use a modified version of the Population, Intervention, Comparator(s), Outcome(s) and Study Design (PICOS) framework to summarise data for extraction.[26] However, due to the inclusion of observational studies in our review, we will replace the term 'intervention' with 'exposure', and 'study design' will be replaced by 'study characteristics'. We will extract information for each component of the PICOS framework, in addition to study results for each study included in the review (table 3).

### Exposures
Our exposures of interest will be risk factors for mental illness in people with atopic eczema or psoriasis. We will consider any variable that authors of included papers have conducted analyses to assess whether they are associated with mental illness in people with atopic eczema or psoriasis as potential risk factors. These may include sociodemographic factors (eg, sex, ethnicity and deprivation), lifestyle factors (eg, level of physical activity, diet and alcohol consumption) or environmental factors.

### Outcomes
Our primary outcome of interest will be mental illness in individuals with atopic eczema or psoriasis. Mental

**Table 3** Items that will be collected using the data extraction form

| Parameter | Information for extraction |
|---|---|
| Population | Participant inclusion and exclusion criteria<br>Demographic characteristics (age, sex and ethnicity distributions)<br>Sample size |
| Exposure | Definition and identification of individuals with the risk factor(s) of interest<br>Number of individuals with the risk factor(s) of interest |
| Comparator | Definition and identification of individuals without the risk factor(s) of interest<br>Number of individuals without the risk factor(s) of interest |
| Outcome | Definition and identification of mental illness outcome(s)<br>Number of individuals in exposed and comparison group with the outcome |
| Study characteristics | Bibliographic information (authors, journal, publication year, volume, page numbers and doi)<br>Study design<br>Study setting<br>Study sampling frame<br>Methods of participant recruitment<br>Aims and objectives |
| Study results | Unadjusted and fully adjusted effect estimates for the association between risk factors and mental illness<br>Confounders measured and adjusted for analysis |

illness will be grouped into two broad categories (CMD or SMI); unless there are sufficient studies looking at specific mental illnesses (eg, depression), then we will also explore by specific mental illness subgroup. We will include studies regardless of how they capture mental illness outcomes (ie, we will include clinical diagnoses or self-reported mental illness established with or without validated tools).

### Risk of bias assessment for individual studies

Two reviewers (EA and YS) will independently assess the risk of bias for the larger of 10%, or five, of the included studies. Any disagreements will be discussed so that a consensus can be reached. A third reviewer (KM) will be available to arbitrate if required. A single reviewer (EA) will then assess risk of bias for the remaining studies. We will use the Quality in Prognosis Studies (QUIPS) tool to assess the risk of bias of included studies.[27] QUIPS assesses and evaluates the risk of bias in six different domains: (1) study participation, (2) study attrition, (3) prognostic factor measurement, (4) outcome measurement, (5) study confounding and (6) statistical analysis and reporting.[27] For each study included, we will assess and categorise the risk of bias for each domain into one of three qualitative categories (low, moderate or high risk

of bias) using the prompting items provided within the tool. We will produce separate risk of bias tables for observational studies and randomised controlled trials (RCTs) along with justifications for the decisions made.

### Data synthesis and meta-bias(es)

We will synthesise our results narratively. We will describe and tabulate the results of the studies included in the review according to the study design (RCT, cohort, case–control or cross-sectional studies), skin disease type (either atopic eczema or psoriasis), risk factor under investigation and outcome measure (either CMD or SMI). We will describe and tabulate the results of the RCTs separately from the results of other studies included in the review. If possible, we will also identify risk factors that are common and distinct between atopic eczema and psoriasis. If at least two studies are sufficiently homogeneous (in terms of study design, study population, risk factor assessed and outcome), we will consider a meta-analysis to pool the effect estimates. We will use the $I^2$ statistic to quantify levels of statistical heterogeneity ($I^2$ of 0%–40% may indicate negligible heterogeneity, 30%–60% may indicate moderate heterogeneity, 50%–90% may indicate substantial heterogeneity and 75%–100% may indicate considerable heterogeneity).[24] If possible, we will also consider meta-regression to investigate whether study characteristics (eg, study design, risk of bias, study outcome and skin disease) or the demographics of the study population (eg, age and sex) are associated with the magnitude of effects and can explain any observed statistical heterogeneity. We will assess the risk of publication bias for the studies included in the review using funnel plots. We will use STATA V.16.0 to perform all statistical analysis.

### Confidence in cumulative evidence

We will use the Grading of Recommendations, Assessment, Development and Evaluation (GRADE) framework to evaluate and summarise the quality of cumulative evidence for each broad outcome (CMD or SMI) and risk factor pair.[28] If more than one study are identified for a specific subtype of a CMD or SMI (such as depression or schizophrenia) and a specific risk factor, we will use GRADE to summarise the quality of evidence for that subtype. We will categorise the strength of evidence into four qualitative categories: 'high', 'moderate', 'low' or 'very low'. The quality of evidence for included studies will be upgraded if there is a large magnitude of effect or a dose-response gradient.[28] The quality of evidence will be rated down if there is a high risk of bias, imprecision in the study estimate, a high probability of publication bias or inconsistent results.[28] We will present the judgements made during this process in a 'Summary of Findings' table.

### Ethics and dissemination

As this study is a systematic review that does not involve human participation, we do not require ethical approval. We will disseminate the results of this review by publishing

in an open access, peer-reviewed journal and presenting at conferences. We will document any important amendments and protocol deviations, along with justifications, and publish them as an appendix in the final review.

**Author affiliations**
¹Department of Non-Communicable Disease Epidemiology, London School of Hygiene & Tropical Medicine, London, UK
²Siaal Research Center for Family Medicine and Primary Care, Faculty of Health Sciences, Ben-Gurion University of the Negev, Beer Sheva, Israel
³Department of Quality Measurements and Research, Clalit Health Services, Tel Aviv, Israel
⁴Division of Psychiatry, University College London, London, UK
⁵Department of Psychological Medicine, Institute of Psychiatry, Psychology, and Neuroscience, King's College London, London, UK
⁶St John's Institute of Dermatology, Guys and St Thomas' Foundation Trust and King's College London, London, UK
⁷Health Data Research UK, London, UK

**Contributors** EA, SL and KM had the original idea for the review. All authors (EA, YS, JH, RM, AM, LR, LS, CHS, SL and KM) were involved in the design of the study. EA wrote the first draft of the protocol. All authors (EA, YS, JH, RM, AM, LR, LS, CHS, SL and KM) contributed to further drafts and approved the final manuscript. Kate Perris peer reviewed the search strategy.

**Funding** EA was funded by a British Skin Foundation (BSF) PhD studentship (Reference: 024/S/18). SL was funded by a Wellcome Trust Senior Clinical Fellowship in Science (Reference: 205039/Z/16/Z).

**Competing interests** None declared.

**Patient consent for publication** Not required.

**Provenance and peer review** Not commissioned; internally peer reviewed.

**ORCID iDs**
Elizabeth I Adesanya http://orcid.org/0000-0002-8912-7520
Rohini Mathur http://orcid.org/0000-0002-3817-8790
Amy R Mulick http://orcid.org/0000-0002-4009-2080
Sinéad M Langan http://orcid.org/0000-0002-7022-7441
Kathryn E Mansfield http://orcid.org/0000-0002-2551-410X

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
