## [Reviewer comments · BMJ Open]

ARTICLE DETAILS

TITLE (PROVISIONAL)	Risk factors for mental illness in adults with atopic eczema or psoriasis: protocol for a systematic review
AUTHORS	Adesanya, Elizabeth; Schonmann, Yochai; Hayes, Joseph; Mathur, Rohini; Mulick, Amy; Rayner, Lauren; Smeeth, Liam; Smith, C; Langan, Sinead; Mansfield, Kathryn

VERSION 1 – REVIEW

REVIEWER	Robert A. Schwartz MD, MH Rutgers New Jersey Medical School
REVIEW RETURNED	20-Mar-2020

GENERAL COMMENTS	The goal of synthesizing the available evidence regarding the risk factors for mental illness in adults with atopic eczema or psoriasis is laudable. The concept of evaluating risk factors for mental illness in adults with atopic eczema or psoriasis is a complex and important one. Skin disorders can create stress, the reaction to which varies and is variable in each individual. I note that stress is preferentially "psychological distress" is mentioned, but the concern is how to measure its level.
--

REVIEWER	Dr. Bárbara Roque Ferreira Department of Dermatology, Centre Hospitalier de Mouscron, Belgium & Eurometropolis Lille - Kortrijk – Tournai, and Centre for Philosophy of Science of the University of Lisbon, Portugal.
REVIEW RETURNED	08-Apr-2020

GENERAL COMMENTS	Congratulations for your protocol. Well-structured systematic reviews on the link between mental disorders and chronic dermatoses are welcome to highlight the relevance of psychodermatology and mental illness in the clinical practice in Dermatology. Furthermore, the analysis of risk factors for mental illness in this context is an additional important aspect that your paper focuses.
--

REVIEWER	Caroline Farmer University of Exeter, UK
REVIEW RETURNED	11-Aug-2020

GENERAL COMMENTS	I am grateful to have had the opportunity to review this protocol for a systematic review of factors associated with an increased risk of mental health problems in people with atopic eczema and psoriasis. This is a review with both clinical and methodological points of interest, which I think would be of interest to those working in this clinical area.
--

The manuscript is clearly written, and generally speaking all the relevant information about the methods that will be used have been provided. I agree with some of the steps they have taken in their approach, such as the extraction of both adjusted and unadjusted data, and the broad way in which they are defining mental illness (although please see my note below about the search strategy). However, I do have a couple of serious concerns about the approach proposed by the authors. These may be resolved through further clarification or justification in the protocol, as perhaps i've misunderstood the aims/methods intended. However, I submit this as a major revision, in case these concerns cannot be easily resolved.

Major points

- The authors refer to this as a review of risk factors; however on the basis of the methods outlined in the manuscript this term would be inappropriate. My understanding from the protocol is that the authors plan to identify analyses comparing people with atopic eczema or psoriasis who have mental illness with those who do not have mental illness to identify differences (in personal or environmental characteristics, for example). The inclusion criteria for the review specify that cross-sectional data is relevant for inclusion, and it is not clear from the protocol if longitudinal data will be sought or is indeed relevant. The definition of a risk factor is a characteristic that is associated with an increased risk of developing an outcome (in this case mental illness); it is therefore prognostic and requires longitudinal evidence to support. That patients with and without mental illness in samples of people with atopic eczema or psoriasis differ in some way is not sufficient evidence to support that this characteristic causes people to be at risk of developing mental illness. The authors also clearly outline in their introduction that the relationship between mental illness and atopic eczema/psoriasis is unclear, and no hypothesis about the direction of effect between these factors and any differing characteristics is presented. Longitudinal data may therefore not be of interest to the authors at this stage. From the methods it rather appears that all or the majority of the data identified for the review will be concerning correlates of mental illness in people with atopic eczema/psoriasis: i.e. factors that are more prevalent in people with eczema/psoriasis with mental illness than those without. Therefore, unless the authors clarify their approach as one where the target evidence is prognostic, I would suggest that the authors use the term 'correlates' rather than 'risk factors' throughout.

- The risk of bias tools chosen to evaluate the quality of RCTs and NRS are inappropriate for the purposes of this study. While these tools are appropriate for characterising the 'trustworthiness' of effect estimates reported in those studies for evaluating the efficacy of interventions, they do not provide an insight into the trustworthiness of effect estimates comparing patient characteristics within the study sample. It would be better to use a risk of bias tool for case control evidence, or else to use QUIPS (for prognostic data, or modified for use with correlates).

Minor issues

- The authors are taking significant steps to identify trials in their search strategy, but this seems odd given that for the aims of this review observational studies (that generally have broader inclusion

	criteria) would be more appropriate. Please can the authors justify their use of trial evidence in this review? Also please can the authors clarify the inclusion criteria for interventional studies; e.g. will comparisons using baseline characteristics only be used, or will you also use post-intervention characteristics? If the latter, are there any exclusion criteria around relevant interventions?  • In Table 1 the authors refer to “risk factors of interest”. Can you please clarify how you will select correlates/risk factors for extraction? Will both population and environmental factors be relevant? Will any data be extracted, regardless of whether there is a statistically significant difference between people with/without mental illness? • The search strategy is generally ok and clearly outlined. It would be helpful if the authors could please specify which databases within WoS they will search, and also for the Cochrane library? The authors state that they wish to have broad inclusion criteria for mental illness, though the terms used for mental illness are not comprehensive (e.g. what about eating disorders, OCD). If too late (as search is underway) the authors may wish to note this as a limitation, or state if they believe this evidence would be picked up using other means. There is some overlap in the terms (e.g. anxiety, anxiety; depression, depression*) but this obviously won't affect the output. Good luck with your review, I look forward to reading your findings!
--	---

VERSION 1 – AUTHOR RESPONSE

Reviewer 1

COMMENT 1.1

The goal of synthesizing the available evidence regarding the risk factors for mental illness in adults with atopic eczema or psoriasis is laudable. The concept of evaluating risk factors for mental illness in adults with atopic eczema or psoriasis is a complex and important one. Skin disorders can create stress, the reaction to which varies and is variable in each individual. I note that stress is preferentially "psychological distress" is mentioned, but the concern is how to measure its level.

RESPONSE 1.1

Thank you for your kind comments. With respect to your concerns regarding ‘stress and psychological distress’, we included ‘psychological distress’ as a keyword in our search strategy as a broad search term to capture studies that did not state a specific mental illness in the fields searched, but nonetheless studied one of the specific mental illness outcomes we aim to include (i.e. depression, anxiety, bipolar disorder, schizophrenia, etc.). We appreciate that stress is a difficult concept to capture, and that it is not the focus of our review, so we do not plan to include studies where the outcome investigated is stress alone.

Reviewer 2

COMMENT 2.1

Congratulations for your protocol. Well-structured systematic reviews on the link between mental disorders and chronic dermatoses are welcome to highlight the relevance of psychodermatology and mental illness in the clinical practice in Dermatology. Furthermore, the analysis of risk factors for mental illness in this context is an additional important aspect that your paper focuses.

RESPONSE 2.1

Thank you for your kind comments.

Reviewer 3

COMMENT 3.1

MAJOR POINTS

The authors refer to this as a review of risk factors; however on the basis of the methods outlined in the manuscript this term would be inappropriate. My understanding from the protocol is that the authors plan to identify analyses comparing people with atopic eczema or psoriasis who have mental illness with those who do not have mental illness to identify differences (in personal or environmental characteristics, for example). The inclusion criteria for the review specify that cross-sectional data is relevant for inclusion, and it is not clear from the protocol if longitudinal data will be sought or is indeed relevant. The definition of a risk factor is a characteristic that is associated with an increased risk of developing an outcome (in this case mental illness); it is therefore prognostic and requires longitudinal evidence to support. That patients with and without mental illness in samples of people with atopic eczema or psoriasis differ in some way is not sufficient evidence to support that this characteristic causes people to be at risk of developing mental illness. The authors also clearly outline in their introduction that the relationship between mental illness and atopic eczema/psoriasis is unclear, and no hypothesis about the direction of effect between these factors and any differing characteristics is presented. Longitudinal data may therefore not be of interest to the authors at this stage. From the methods it rather appears that all or the majority of the data identified for the review will be concerning correlates of mental illness in people with atopic eczema/psoriasis: i.e. factors that are more prevalent in people with eczema/psoriasis with mental illness than those without. Therefore, unless the authors clarify their approach as one where the target evidence is prognostic, I would suggest that the authors use the term 'correlates' rather than 'risk factors' throughout.

RESPONSE 3.1

Thank you, we appreciate your detailed consideration of our methodology. Your comments have highlighted some potential confusion regarding the aim of our systematic review. Using a broad definition, we are regarding 'risk factors' as variables associated with an increased risk of a disease – any link does not need to be causal. For example, the incidence of eczema differs by age (i.e. the condition peaks between the ages of two and four). Age does not cause eczema, but, for the purposes of our review, we would regard age as a risk factor for eczema. Our aim with this review is to identify 'risk factors' for mental illness in people with skin disease. So, we will be looking at studies of people with skin disease and comparing those with and without a specific risk factor to identify whether they have mental illness or not. Here, longitudinal data is preferred as it will allow us to

address whether there is a temporal relationship between potential 'risk factors' and mental illness in people with skin disease, and potentially provide evidence suggesting a causal link between each 'risk factor' and mental illness pair. However, other non-longitudinal study designs (i.e. cross-sectional or case-control studies) will also provide useful information that may help us identify characteristics associated with an increased risk of mental illness in people with skin disease (such as the relationship between age and eczema).

Studies will only be eligible for inclusion in the review if an effect estimate of the risk factors for mental illness in adults with atopic eczema or psoriasis are reported within the study. This means we will only include studies where ratio measures (i.e. odds, risk or hazard ratios) or difference measures (i.e. mean differences or standardised mean differences) have been reported. We will not include studies which only report correlates of mental illness in people with atopic eczema or psoriasis. We have added some additional text to the manuscript and eligibility criteria of our manuscript to clarify our methodology.

MANUSCRIPT CHANGES 3.1

Location – Introduction, page 4

The temporal sequence of the associations between skin disease and mental illness are also well recognised, with evidence suggesting that atopic eczema or psoriasis precede mental illness diagnosis.^{10,12}

Location – Introduction, page 5

In the context of this systematic review, we will use the term 'risk factor' to refer to variables associated with an increased risk of mental illness in individuals with atopic eczema or psoriasis.

Location – Table 1: Eligibility criteria (Inclusion criteria), page 7

All RCTs, cohort, case-control and cross-sectional studies where an effect estimate (i.e. ratio or difference measures) of the risk factors for mental illness in adults with atopic eczema or psoriasis are reported

Location – Table 1: Eligibility criteria (Exclusion criteria), page 7

Studies where correlates (without a measure of effect) have been calculated to estimate the association between a risk factor and mental illness in adults with atopic eczema or psoriasis.

COMMENT 3.2

The risk of bias tools chosen to evaluate the quality of RCTs and NRS are inappropriate for the purposes of this study. While these tools are appropriate for characterising the 'trustworthiness' of effect estimates reported in those studies for evaluating the efficacy of interventions, they do not provide an insight into the trustworthiness of effect estimates comparing patient characteristics within the study sample. It would be better to use a risk of bias tool for case control evidence, or else to use QUIPS (for prognostic data, or modified for use with correlates).

RESPONSE 3.2

Thank you for your comments. We agree that ROBINS-I and RoB 2 are not ideal for this study as both tools evaluate bias of effect estimates in studies with interventions. Thank you for your

recommendation of the Quality in Prognosis Studies (QUIPS) tool, we now plan to use it to assess the risk of bias of included studies. We have edited the manuscript appropriately.

MANUSCRIPT CHANGES 3.2

Location – Abstract: Methods and analysis, page 2

We will assess the risk of bias of included studies using the Quality in Prognosis Studies (QUIPS) tool.

Location – Methods: Risk of bias assessment for individual studies, page 12

We will use the Quality in Prognosis Studies (QUIPS) tool to assess the risk of bias of included studies.²⁷ QUIPS assesses and evaluates the risk of bias in six different domains: (1) study participation; (2) study attrition; (3) prognostic factor measurement; (4) outcome measurement; (5) study confounding; and (6) statistical analysis and reporting.²⁷ For each study included, we will assess and categorise the risk of bias for each domain into one of three qualitative categories (low, moderate or high risk of bias) using the prompting items provided within the tool.

COMMENT 3.3

MINOR COMMENTS

The authors are taking significant steps to identify trials in their search strategy, but this seems odd given that for the aims of this review observational studies (that generally have broader inclusion criteria) would be more appropriate. Please can the authors justify their use of trial evidence in this review? Also please can the authors clarify the inclusion criteria for interventional studies; e.g. will comparisons using baseline characteristics only be used, or will you also use post-intervention characteristics? If the latter, are there any exclusion criteria around relevant interventions?

RESPONSE 3.3

Thank you, the inclusion of trials was a topic of considerable discussion when we developed our protocol. We decided that it was important to identify relevant trials for inclusion as it will allow us to investigate 'untreated skin disease' as a risk factor of mental illness in individuals with atopic eczema or psoriasis. We plan to compare the change in mental illness outcomes from baseline measurements to post-intervention measurements (between groups receiving the intervention to groups receiving no intervention) to see the effect that leaving skin disease untreated has on mental illness in people with atopic eczema or psoriasis. We will report the results of trials included in the systematic review separately from the results of other studies. We have added some additional text to the manuscript to clarify our approach.

MANUSCRIPT CHANGES 3.3

Location – Methods: Data synthesis and meta-bias(es), page 13

We will describe and tabulate the results of the randomised controlled trials separately from the results of other studies included in the review.

COMMENT 3.4

In Table 1 the authors refer to “risk factors of interest”. Can you please clarify how you will select correlates/risk factors for extraction? Will both population and environmental factors be relevant? Will any data be extracted, regardless of whether there is a statistically significant difference between people with/without mental illness?

RESPONSE 3.4

Thank you for your comments. We will classify any variable that authors of included papers have performed statistical analysis to assess whether they are associated with mental illness in people with atopic eczema or psoriasis as potential risk factors. Hence, we will include both sociodemographic, anthropometric and environmental factors provided these have been explored as factors associated with mental illness in those with psoriasis or atopic eczema. We acknowledge that this approach may mean that some of the studies included may not have been designed to analyse the specific variables we identify as ‘risk factors’ as exposures of interest, and so may not completely capture all relevant confounders in adjusted models. We will take incomplete capture of confounders into account when presenting the findings of the included studies. We will extract data on the risk factors identified, regardless of whether there is a statistically significant measure of effect for the association between the variable and mental illness, as null findings will still offer insight into the relationships between each risk factor/outcome pair in people with atopic eczema or psoriasis. We have added some additional text to the manuscript to clarify our approach.

MANUSCRIPT CHANGES 3.4

Location – Methods: Exposures, page 12

Our exposures of interest will be risk factors for mental illness in people with atopic eczema or psoriasis. We will consider any variable that authors of included papers have conducted analyses to assess whether they are associated with mental illness in people with atopic eczema or psoriasis as potential risk factors. These may include sociodemographic factors (e.g. sex, ethnicity, deprivation), lifestyle factors (e.g. level of physical activity, diet, alcohol consumption) or environmental factors.

COMMENT 3.5

The search strategy is generally ok and clearly outlined. It would be helpful if the authors could please specify which databases within WoS they will search, and also for the Cochrane library?

RESPONSE 3.5

Thank you for your comments. In Web of Science, we will search the Web of Science Core Collection, which includes: the Science Citation Index Expanded (SCI-EXPANDED); the Social Sciences Citation Index (SSCI); the Arts & Humanities Citation Index (A&HCI); the Conference Proceedings Citation Index-Science (CPCI-S); the Conference Proceedings Citation Index – Social Science & Humanities (CPCI-SSH); and the Emerging Sources Citation Index (ESCI). In the Cochrane Library, we will search Cochrane Reviews, Cochrane Protocols, Trials, Editorials, Special Collections, Clinical Answers and Other Reviews. We have included some additional text in the manuscript to clarify this approach.

MANUSCRIPT CHANGES 3.5

Location – Methods: Information sources, page 8

We will search the following databases for relevant articles from inception to March 2020: Medline, Embase, Global Health, Scopus, Cochrane Library (which includes Cochrane Reviews, Cochrane Protocols, Trials, Editorials, Special Collections, Clinical Answers and Other Reviews), Web of Science (which includes the Science Citation Index Expanded [SCI-EXPANDED]; the Social Sciences Citation Index [SSCI]; the Arts & Humanities Citation Index [A&HCI]; the Conference Proceedings Citation Index-Science [CPCI-S]; the Conference Proceedings Citation Index – Social Science & Humanities [CPCI-SSH]; and the Emerging Sources Citation Index [ESCI]), Base, PsycInfo and the Global Resource of Eczema Trials (GREAT).

COMMENT 3.6

The authors state that they wish to have broad inclusion criteria for mental illness, though the terms used for mental illness are not comprehensive (e.g. what about eating disorders, OCD). If too late (as search is underway) the authors may wish to note this as a limitation, or state if they believe this evidence would be picked up using other means. There is some overlap in the terms (e.g. anxiety, anxi*; depression, depress*) but this obviously won't affect the output.

RESPONSE 3.6

We chose to focus our study on risk factors for common mental disorders (i.e. depression, anxiety) and severe mental illnesses (i.e. schizophrenia, bipolar disorder and other psychoses). We agree that exploring the link with other mental illnesses, including eating disorders and obsessive-compulsive disorder, would be interesting, but we feel it is beyond the scope of this review. We agree that there is some overlap between the mental illnesses terms that we have included in our search terms and have edited our search strategy appropriately.

MANUSCRIPT CHANGES 3.6

Location – Table 2: Keywords included in the search strategy for all databases

Location – Supplementary Table 1: Search strategy in MEDLINE database

VERSION 2 – REVIEW

REVIEWER	Bárbara Roque Ferreira Belgium
REVIEW RETURNED	14-Oct-2020

GENERAL COMMENTS	Congratulations for this paper and protocol, which focuses on a relevant topic in psychodermatology.
--

REVIEWER	Caroline Farmer University of Exeter, UK
REVIEW RETURNED	06-Oct-2020

GENERAL COMMENTS	Many thanks for your considered review and response of my comments. I really like the changes that you've made to the protocol and the explanatory text, and i think they adequately address the issues i raised. I had an outstanding query about whether evaluating change in outcomes in untreated eczema/psoriasis fits within the aims of this review, as i'm not sure being untreated can be usefully defined as a risk factor for a disease...? Are you not then evaluating the efficacy of the intervention? Possibly I have misunderstood, and I imagine you will explore all this in your results anyway, so I didn't think it necessary for you to change anything in the protocol. Good luck with the review! I look forward to reading your findings (and continuing to debate your use of the term 'risk factor' :))!
---